# Curing Kinetics of Methylene Diphenyl Diisocyanate—Based Polyurethane Elastomers

**DOI:** 10.3390/polym14173525

**Published:** 2022-08-27

**Authors:** Shuang Liu, Xiaodong Li, Mengchen Ge, Xujie Du, Meishuai Zou

**Affiliations:** 1School of Materials Science and Engineering, Beijing Institute of Technology, Beijing 100081, China; 2System Design Institute of Hubei Aerospace Technology Academy, Wuhan 432000, China

**Keywords:** differential scanning calorimetry, polyurethane, diphenylmethane diisocyanate, curing reaction kinetics

## Abstract

The curing kinetics of MDI-based polyurethane elastomers were studied by non-isothermal differential scanning calorimetry (DSC). The kinetic parameters of the reaction system were calculated by the Kissinger method. The changing activation energy was observed by the Flynn–Wall–Ozawa method and the Friedman method. The results of model free fitting showed that the curing reaction could be divided into two stages, showing a change in reaction order when α > 0.45 and a piecewise curing mechanism function of the MDI-based polyurethane elastomers reaction system was deduced by autocatalytic model. The extrapolation method was used to determine the optimum curing conditions for the system, which can accurately describe the curing process. In addition, the optimal curing conditions are when: the constant temperature curing temperature of the system is 81 °C, the curing time is 29 min, and the post-curing temperature is 203 °C.

## 1. Introduction

Polyurethane (PU) is a class of multi-purpose synthetic resins with various product forms [1]. Depending on its chemical structure and preparation process, PU can be used as elastomer, adhesive, coating, etc. [2,3,4]. PU contains carbamate group (-NH-COO-) in its molecular structure, which is generally obtained by the reaction of isocyanate and polyol. It has a special micro-phase separation structure of soft and hard segments and exhibits excellent damping properties [5,6]. Therefore, polyurethane elastomers are often used as vibration-damping and sound-absorbing materials in various scenarios, such as train seats, vibration-damping bearings for high-speed trains, etc. [7]. Compared with traditional polyurethane based on Toluene diisocyanate (TDI), polyurethane-based on Methylene diphenyl diisocyanate (MDI) has many advantages. The molecular structure of MDI has good symmetry and better flexibility, so the polyurethane molecular chain is more regular and easier to crystallize. MDI-based polyurethane has better heat resistance and mechanical properties than TDI-based polyurethane. Moreover, MDI is less toxic, cheaper and less harmful to the human body and environment; using MDI instead of TDI to synthesize polyurethane is more conducive to environmental protection [8,9].

Curing kinetics studies can explain better the reaction process and provide theoretical guidance for the regulation of the curing reaction. The curing kinetics parameters and curing conditions can have a direct impact on the structure and properties of polyurethane. Therefore, analyzing the curing kinetics of polyurethane materials is essential to precisely monitor and control the curing of polyurethane. Common methods used to study curing reactions include DSC [10], FTIR [11], NMR [12], rheological [13], and chromatographic techniques [14]. There have been some studies on the curing kinetics of polyurethane. Adrian Olejnik et al. investigated the curing kinetics of a composite polyurethane system by DSC method, and the results showed that the system is a complex reaction [15]. The kinetic parameters of IPDI-type polyurethanes were observed and calculated by Rodrigues et al. [16]. Fernandez et al. utilized iso-conversional methods to describe the kinetics of the curing reaction of HDI-type polyurethanes under isothermal conditions [17]. However, most of the previous studies were focused on IPDI, TDI and other types of polyurethanes, and there are few studies on the curing kinetics of MDI-based polyurethane. Different from other isocyanates, the curing reaction of MDI-based polyurethane is a fast curing process, making it harder to conduct kinetic analysis. As a widely used raw material for the polyurethane industry, MDI has a significant presence in the market, so understanding the curing behavior of MDI-based polyurethane is of great value for optimizing industrial production.

In this work, the curing kinetics of MDI-based polyurethanes were studied by non-isothermal DSC test to investigate the curing reaction of MDI with polyether polyols. The curing kinetic parameters and kinetic mechanism were determined. Furthermore, the theoretical guidance for the curing process of MDI-based polyurethanes was provided.

## 2. Materials and Methods

### 2.1. Materials

Methylene diphenyl diisocyanate (MDI), Polyether Polyol (330 N-4950) and polytetrahydrofuran glycol (PTMG-2000) were purchased from Wanhua Chemical Group Co., Ltd., Shandong, China. Chain extender 1,4-butanediol (BDO) was obtained from Shanghai Macklin Biochemical Technology Co., Ltd., Shanghai, China. BDO was analytically pure, and the others were chemically pure.

### 2.2. Preparation of Polyurethane Curing Systems

Pre-treatment of raw materials was needed. The raw materials 330 N, PTMG-2000 and BDO were vacuum-dehydrated at 105 °C~110 °C for 2 h and then cooled to room temperature before use.

Component A: measured dehydrated PTMG-2000, 330 N and BDO were added in a three-mouth flask and mixed well for 1 h at a constant temperature of 100~115 °C under vacuum. Component B: the dehydration reaction of PTMG-2000 and MDI was conducted under 80~85 °C for 4 h to produce prepolymer with an isocyanate index of 12.50%. For thermal characterization, a certain amount of component A and component B were mixed and kept properly, and a sample was taken from it for thermal characterization.

### 2.3. Differential Scanning Calorimetry

Differential Scanning Calorimetry analysis was performed using a differential scanning calorimeter (DSC 204F1, (Netzsch Co. Ltd., Freistaat, Germany).

The measurement conditions were as follows: sample weight 5–10 mg, nitrogen gas flow-20 mL/min, aluminum pan. Samples were heated from −15 °C to 230 °C at different rates (5, 10, 15 and 20 K/min).

### 2.4. Fourier Transform-Infrared Analysis

FTIR spectroscopy (Nicolet 6700, Thermo Fisher Scientific, Massachusetts, USA) was used to analyze the prepared prepolymer and reacted polyurethanes after DSC tests. With air as the background, the samples were tested in the attenuated total reflection mode. The resolution of the instrument was 4 cm^−1^, and each sample was scanned 32 times with the wavelength range from 4000 cm^−1^ to 450 cm^−1^.

## 3. Theoretical Basis of Curing Kinetics

For common curing reactions, the degree of conversion(*α*) of the curing reaction and the reaction rate can be expressed by Equations (1) and (2), respectively [18]:(1)$$\alpha =\frac{\Delta H_{t}}{\Delta H_{total}}$$
(2)$$\frac{d\alpha }{dt}=\frac{d\alpha }{dT}=k\left(T\right)f\left(\alpha \right)$$
where *α* is the degree of conversion, Δ*H_t_* is the enthalpy of the reaction in certain temperature ranges and time conditions, Δ*H_total_* is the total enthalpy of the curing reaction, *t* is the reaction time, *k*(*T*) is the reaction rate constant, and it only depends on temperature. The *k*(*T*) is given by the Arrhenius equation:(3)$$k\left(T\right)=Aexp\left(-\frac{E_{a}}{RT}\right)$$
where *A* is the pre-exponential factor, *E*_a_ is the apparent activation energy of the reaction, R is the universal gas constant [8.3145 J·mol^−1^·K^−1^], and *T* is the absolute temperature. Substituting Equation (3) into Equation (2) and substituting the rate of heating *β* = *dT*/*dt*, the reaction rate equation should be:(4)$$\frac{d\alpha }{dt}=\frac{d\alpha }{dT}=\frac{A}{\beta }exp\left(-\frac{E_{a}}{RT}\right)f\left(\alpha \right)$$

The purpose of the curing kinetics study is to obtain the “kinetic triple factors”, namely *E*, A and *f*(α) in the above equation of the curing reaction, so as to obtain the MDI-based polyurethane curing reaction characteristics and provide a theoretical basis for practical application.

Kissinger method is commonly used to study the kinetics of curing reactions, which assumes that the conversion at each DSC exothermic peak is invariant and independent of the heating rates [19]. Using the different temperatures corresponding to the peaks of the DSC curves at multiple heating rates, we can calculate the activation energy (*E_a_*) without assuming any kinetic parameter model and integrating the exothermic peak. The Kissinger equation is [20]:(5)$$ln\frac{\beta }{T_{P}^{2}}=\ln\left(\frac{AR}{E_{a}}\right)-\frac{E_{a}}{RT_{p}}$$
where *β* is the heating rate, R is the universal gas constant, *T_p_* is the peak temperature of the DSC curve, *E_a_* is the apparent activation energy, and *A* is the pre-exponential factor.

### 3.1. Isoconversional Methods: Model-Free Kinetics (MFK)

The Kissinger method can only give a single *E_a_* value that represents the average level of the reaction. However, the *E_a_* value varies with the curing degree, which cannot be observed by the Kissinger method. The model-free methods were based on the assumption that both activation energy and pre-exponential factor are functions of conversion, which can help explain the mechanism at different stages of the reaction by calculating the kinetic parameters at different conversion rates. To gain a deeper insight into the curing reaction of MDI-based polyurethanes, the Flynn–Wall–Ozawa method and the Friedman method were considered to investigate the relationship between the activation energy and conversion.

#### 3.1.1. Flynn-Wall-Ozawa Method

The temperatures obtained from the DSC curves at different conversions and different heating rates were used to calculate the activation energy at different curing degrees with the following equation [21,22,23]:(6)$$lg\beta =lg\left(\frac{AE}{RG\left(\alpha \right)}\right)-2.315-0.4567\frac{E}{RT}$$

For different conversions, *A*′ can be expressed as:(7)$$A^{\prime }=lg\left(\frac{AE}{RG\left(\alpha \right)}\right)-2.315$$
where *α* is the degree of conversion, *G*(*α*) is the integral conversion function, and *T* is the temperature at the corresponding conversion. A plot of *lgβ* as a function of 1/*T* using various heating rates yields straight lines. For the curing system, the slope of the regression curve at each conversion rate is used to obtain *E_a_* and *A*′ at corresponding conversion degree.

#### 3.1.2. Freidman Method

Similar to the FWO method, the Friedman method is also a non-model method; it uses the rate of change in the conversions obtained from the DSC curve at different heating rates at a certain conversion rate to calculate the activation energy, and its empirical equation is [24]:(8)$$ln\left(\frac{d\alpha }{dt}\right)=lnA+nln\left(1-\alpha \right)-\frac{E}{RT}$$

For different conversion rates, *A′* can be expressed as:(9)$$A^{\prime }=lnA+nln\left(1-\alpha \right)$$
where *n* is the reaction order, dα/dt is the rate of change in the conversion. When the conversion α is certain, a straight line can be obtained by plotting *ln*(*d**α*/*d**t*) − 1/*T*. The activation energy *E_a_* can then be calculated from the slope.

## 4. Results

### 4.1. Curing Progress Analysis of MDI-Based Polyurethane System

To demonstrate the curing progress of MDI-based polyurethane, it was subjected to non-isothermal DSC tests at different heating rates, and the results are shown in Figure 1. It can be seen from Figure 1 that there is an obvious melting heat absorption peak during the heating process, and as the heating rate increases, the melting heat absorption peak becomes sharper. When the heating rate is lower, there are two peaks for melting. However, when increasing the heating rates, the peaks merge together because the melting temperatures of these two components are slightly different. The resolution and sensitivity of the DSC became lower when the heating rate increased, which resulted in the coalescence of two peaks under high heating rates.

In addition, the characteristic temperatures of the curing reaction of the system at different heating rates can be obtained from the curves in Figure 1, including the onset temperature *T_i_*, the peak temperature *T_p_*, and the terminal temperature *T_f_*. As can be seen from Figure 1 and Table 1, with the increase in the heating rate, the onset temperature *T_i_*, the peak temperature *T_p_*, and the terminal temperature *T_f_* of the system gradually increased, while the curve shifted to high temperature. Moreover, the area of exothermic peak is increased. It can be explained because the curing reaction occurs more easily and quickly with increasing the heating rate. Therefore, the thermal effect per unit time increases and the heat generated per unit time increases so that the peak shifts to a higher value [25].

According to Equation (1), the variation trend of curing degree with temperature at different heating rates can be obtained by integrating the DSC curve in Figure 1. As shown in Figure 2, the curing degree increases abruptly in a short period of time, which may be a result of an autocatalytic effect in the polyurethane curing reaction.

A linear relationship was obtained by fitting ln(*β*/*T_p_^2^*) to 1/*T_p_* in Figure 3. The values of *E_a_* and *A* can be obtained by the Kissinger method in Figure 3, which gives the activation energy *E_a_* = 46.34 kJ·mol^-1^, the pre-exponential factor *A* = 1.01 × 10^6^ min^−1^, and the correlation coefficient R^2^ = 0.9812.

### 4.2. FTIR Analysis

The FTIR spectrums of prepolymer and reacted polyurethane in DSC testing are shown in Figure 4, and the infrared characteristic absorption of the polyurethane molecules can be observed. The peak at 3300 cm^−1^ was due to N-H stretching vibration. The peaks at 1730 cm^−1^ and 1703 cm^−1^ were characteristic absorption peaks of C-N-H bending vibration [26]. The peak at 1370 cm-1 was attributed to the symmetrical deformation vibration of the methyl groups. The peak at 2250 cm^−1^ was a characteristic absorption peak for -NCO. As shown in Figure 4, almost no isocyanate signal can be observed after the reaction, while a strong signal was shown in the spectrum of the prepolymer. This indicates that the reaction was fairly complete in DSC tests.

### 4.3. Model Free Kinetic Models

The relationship between curing degree and *E_a_* values can be obtained by model free kinetic analysis. The change in *E_a_* values in MDI-based polyurethane system was studied by the Flynn–Wall–Ozawa method and the Friedman method, and the results are shown in Figure 5 and Figure 6.

As can be seen, both methods show similar changing trends of *E_a_* value in Figure 6. In the MDI-based polyurethane curing reaction process, the apparent activation energy gradually decreases with the progress of the reaction. The activation energy drops sharply when 0 < *α* < 0.45, while the activation energy decreases at a slower rate at α > 0.45. An interpretation of this behavior is that there is an apparent increase in the movements of segments when 0 < *α* < 0.45. With the temperature getting higher, the viscosity of the system becomes lower making it easier for material diffusion [16]. In addition, there is also an autocatalytic effect in the polyurethane curing reaction. As the reaction proceeds, the autocatalytic effect becomes more obvious, thus the apparent activation energy gradually decreases. This phenomenon is consistent with the findings of Fernandez et al. who reported that the formation of urethane groups brought an autocatalytic effect to the reaction system [17]. However, the decrease in *E_a_* value became slower when *α* > 0.45, which can be explained by the increase in crosslinking rate. The mobility of chains was limited by the crosslinking polymer network, and the autocatalytic effect was reduced.

### 4.4. Model Fitting Kinetic Methods

#### 4.4.1. nth Order Model

To further study the curing behavior of the system, the n-order curing reaction model is used. The kinetic parameter values of the MDI-based polyurethane curing reaction obtained by Kissinger method were used in the Crane equation, which can calculate the reaction order of the MDI-based polyurethane reaction [27].
(10)$$f\left(\alpha \right)={\left(1-\alpha \right)}^{n}$$
(11)$$\frac{d\left(ln\beta \right)}{d\left(\frac{1}{T_{p}}\right)}=-\left(\frac{E_{a}}{nR}+2T_{p}\right)$$

When $E_{a}/nR\gg 2T_{p}$, Equation (11) can be changed to Equation (12):(12)$$\frac{d\left(ln\beta \right)}{d\left(\frac{1}{T_{p}}\right)}=-\frac{E_{a}}{nR}$$

As shown in Figure 7, the fitted straight line is obtained by plotting *lnβ* against 1/*T_p_*, and *n* = 0.88 is obtained by substituting the slope of the fitted straight line into Equation (12). From the obtained parameters, the curing mechanism function Equation (13) can be obtained:(13)$$f\left(\alpha \right)={\left(1\;-\;a\right)}^{0.88}$$

Since n is not an integer, it can be determined that the curing reaction is not a simple primary reaction.

#### 4.4.2. Autocatalytic Reaction Model

The nth order curing reaction model is the simplest model to describe the curing behavior of thermosetting plastics. However, based on the non-model kinetic analysis and the results derived from the Crane equation, the autocatalytic model is more applicable to this MDI-type polyurethane curing reaction than the n-order reaction model. Moreover, the curing kinetics can be mainly divided into two stages. To accurately describe the kinetic behavior of the reaction system, the autocatalytic model was used, and piecewise model fitting was utilized for more accurate results. The equation of the autocatalytic model can be expressed as [28,29]:(14)$$f\left(\alpha \right)=\alpha^{m}{\left(1-\alpha \right)}^{n}$$
where *m*, and *n* are the reaction orders of the curing reaction, the kinetic equation of the curing reaction is:(15)$$\frac{d\alpha }{dt}=Aexp\left(-\frac{E_{a}}{RT}\right)\;\alpha^{m}{\left(1-\alpha \right)}^{n}$$

Taking the logarithm of both sides of the equation:(16)$$ln\left(\frac{d\alpha }{dt}\right)=-\frac{E_{a}}{RT}+mln\alpha +nln\left(1-\alpha \right)+lnA$$

The kinetic parameter values of the polyurethane curing reaction of the MDI system obtained by the Kissinger method were used in Equation (16) to calculate *m* and *n*. By performing a multiple linear regression to *ln*(*dα*/*dt*), *lnα* and *ln*(1 − *α*), the parameters can be determined for different heating rates as shown in Table 2. The fitting result is shown in Figure 8. Based on the obtained kinetic parameters, the kinetic equation of the system can be obtained as:(17)$$\frac{d\alpha }{dt}=e^{14.13}\times e^{\left(-\frac{5573}{T}\right)}\;\alpha^{0.3503}{\left(1-\alpha \right)}^{2.1193\;}(\alpha <0.45)$$
(18)$$\frac{d\alpha }{dt}=e^{12.79}\times e^{\left(-\frac{5573}{T}\right)}\;\alpha^{1.3949}{\left(1-\alpha \right)}^{0.9649\;}(\alpha >0.45)$$

### 4.5. Determination of Optimal Curing Conditions

It can be seen in Figure 1 that the curing temperatures and the position of the exothermic peak of the MDI-based polyurethane system were different with the increased heating rate, which makes it difficult to determine the curing temperature of the MDI-based polyurethane in actual curing process. To obtain accurate and reliable characteristic temperatures, the *T*-*β* extrapolation method was used to obtain the curing temperatures when the heating rate of *β* is 0 K/min. As shown in Figure 9, the onset temperature *T_i_*, the peak temperature *T_p_*, and the terminal temperature *T_f_* were extrapolated for *β* = 0, and the optimal cure temperature can be obtained [30]. The onset temperature was 19.41 °C, the peak temperature was 80.9 °C, and the terminal temperature was 203.3 °C.

After integrating Equations (17) and (18), the *α*-*t* relationship is calculated in Figure 10 (the details of calculation are shown in Appendix A). To obtain a curing degree of 99%, the system should be cured under 81 °C for 28.4 min. Considering the actual experimental situation and material diffusion and other factors, the optimal curing conditions for MDI-based polyurethane curing reaction are: the curing reaction starts at 20 °C, then keeps heating up to 81 °C and curing at that temperature for 29 min, and finally heats up to 203 °C for post-curing treatment.

## 5. Conclusions

In this study, the curing kinetics and curing behavior of the MDI-based polyurethane were investigated by the non-isothermal DSC method. Apparent activation energy and pre-exponential factor were calculated by the Kissinger method: *E_a_* = 46.34 kJ·mol^−1^, *A* = 1.01 × 10^6^ min^−1^. The change in activation energy with curing degree was studied by the Flynn–Wall–Ozawa method and the Friedman method. According to the results obtained from model free kinetic analysis, the *E_a_* value decreased as the degree of cure increased, and the process can be divided into two stages. Moreover, the reaction order calculated by the Crane equation was not an integer. Both results led to the conclusion that the autocatalytic model was more suitable for the curing kinetics of MDI-based polyurethane compared with the nth model. Based on the autocatalytic model, a piecewise model was finally applied to derive the kinetic equation for the reaction. Based on the results of DSC scanning curves, the extrapolation method and the kinetic equation, the best cure condition of the MDI-based polyurethane was when the curing reaction starts at 20 °C, then continues to heat up to 81 °C and cures at 81 °C for 29 min, and finally heats up to 203 °C for post-curing treatment.

## Figures and Tables

**Figure 1 polymers-14-03525-f001:**
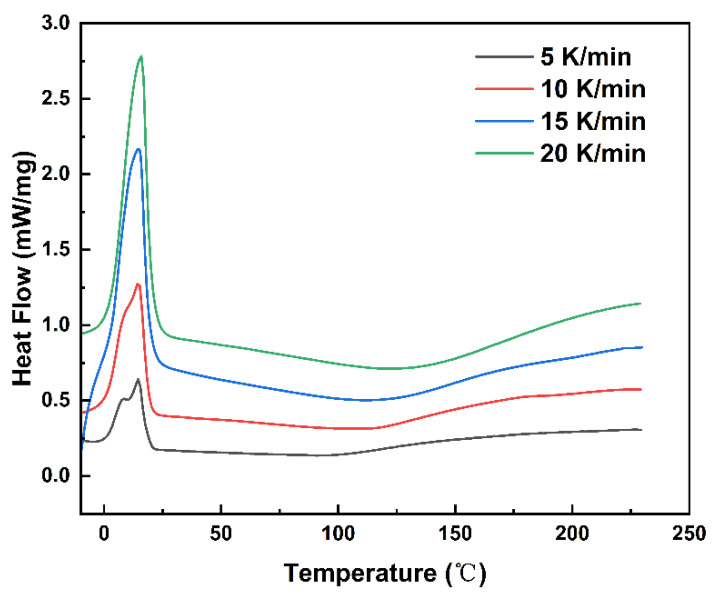
DSC results of the MDI-based polyurethane at different heating rates.

**Figure 2 polymers-14-03525-f002:**
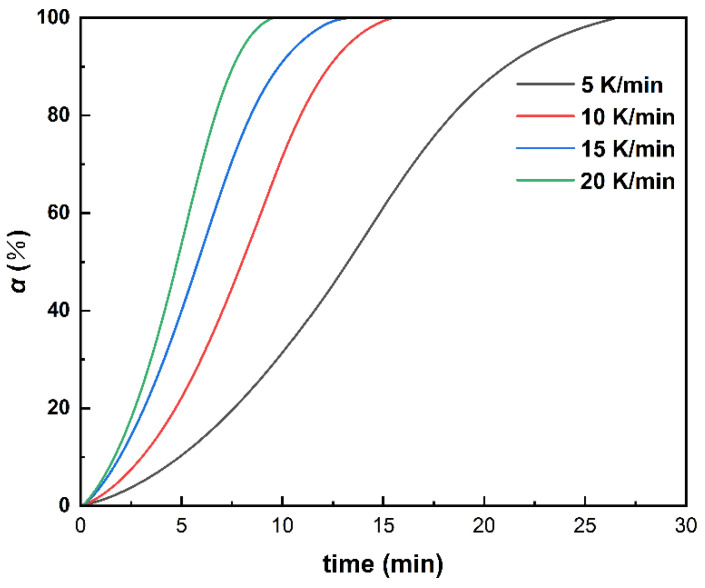
Relationship of the curing degree and time under different heating rates.

**Figure 3 polymers-14-03525-f003:**
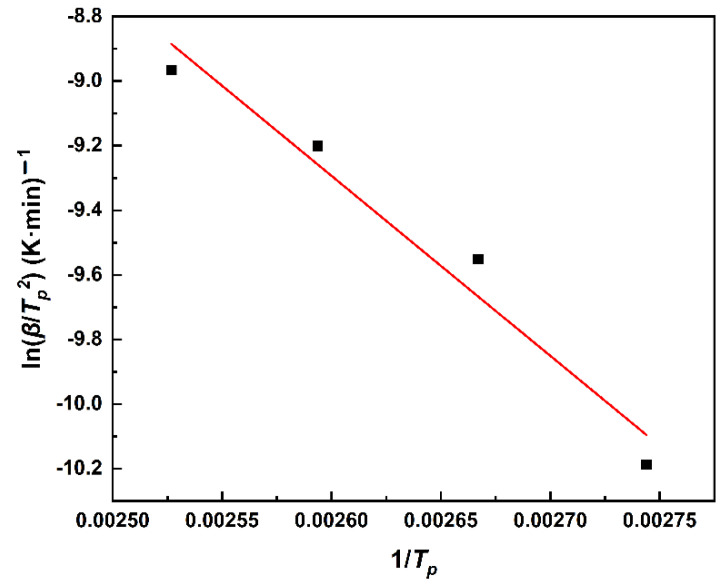
The linear fit of ln*β*/*T_p_^2^* and1/*T*_p_ by the Kissinger method.

**Figure 4 polymers-14-03525-f004:**
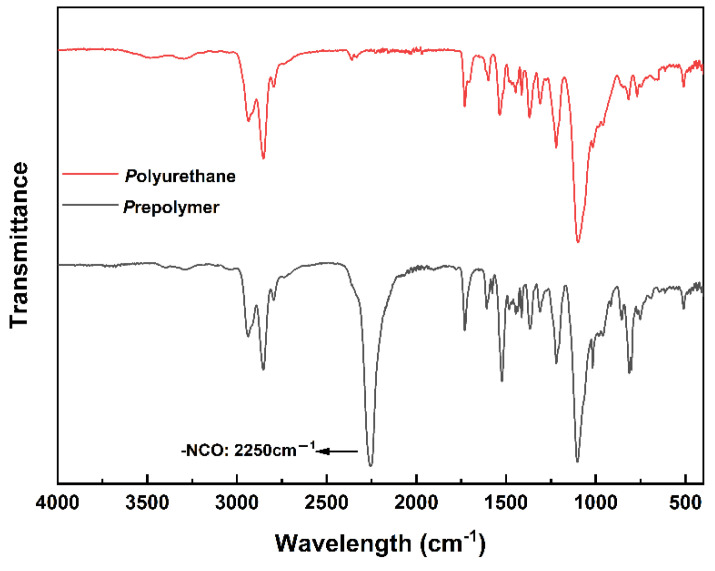
FTIR spectrum of prepolymer and reacted polyurethane after DSC testing.

**Figure 5 polymers-14-03525-f005:**
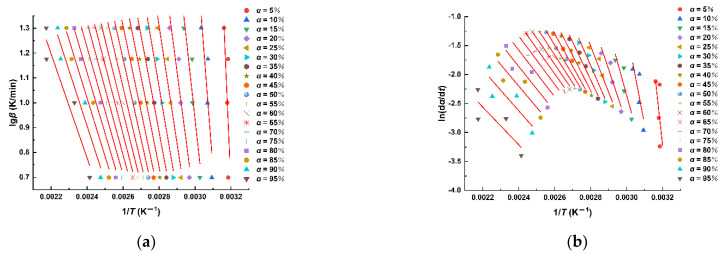
(**a**) The Flynn–Wall–Ozawa analysis of MDI-based polyurethane; (**b**) The Friedman analysis of MDI-based polyurethane.

**Figure 6 polymers-14-03525-f006:**
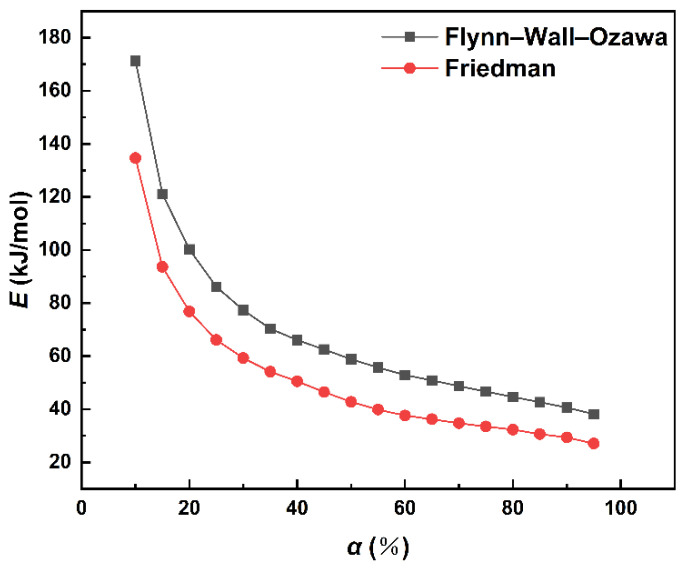
Dependence of the *E_a_* value on the curing degree.

**Figure 7 polymers-14-03525-f007:**
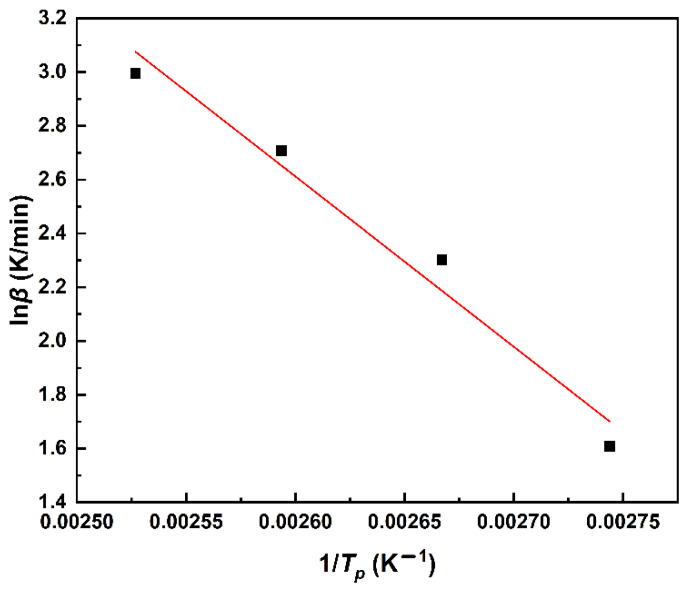
The linear fit of ln*β* and 1/*T*_p_ by Crane equation.

**Figure 8 polymers-14-03525-f008:**
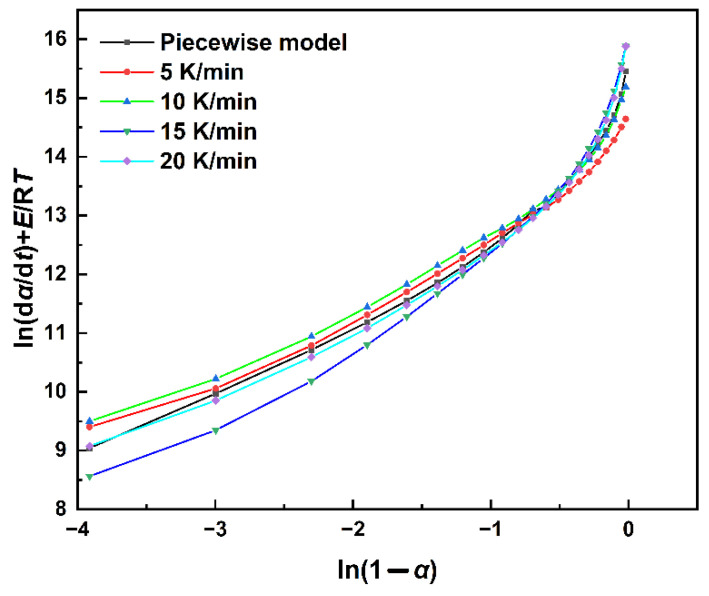
Model fitting results using the piecewise autocatalytic model.

**Figure 9 polymers-14-03525-f009:**
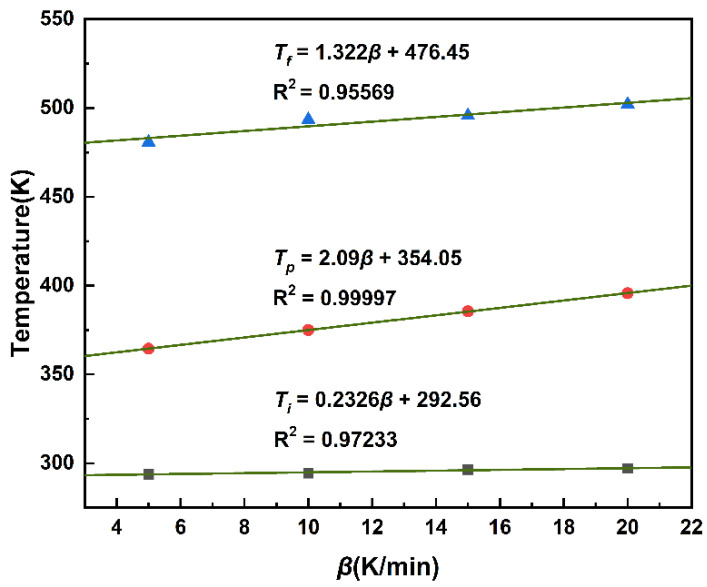
Characteristic curing temperatures.

**Figure 10 polymers-14-03525-f010:**
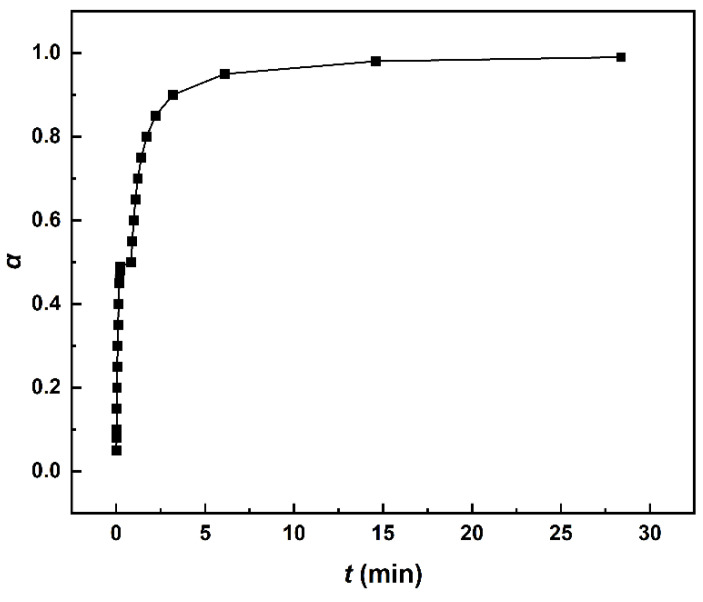
The curve of *α*-*t* under optimal curing conditions.

**Table 1 polymers-14-03525-t001:** Characteristic temperatures and exothermic enthalpy of the MDI-based polyurethane at different heating rates.

*β*/K·min^−1^	*T_i_*/°C	*T_p_*/°C	*T_f_*/°C	Δ*H*/J·g^−1^
5	20.7	91.3	207.6	84.3
10	21.3	101.8	220.3	100.5
15	23.3	112.4	222.5	117.5
20	23.9	122.6	228.9	134.2

**Table 2 polymers-14-03525-t002:** Kinetic parameters of MDI-based polyurethane curing reactions calculated by Autocatalytic model.

*β*/K·min^−1^	*α* < 0.45				*α* > 0.45			
	*m*	*n*	*A* × 10^6^	*R* ^2^	*m*	*n*	*A* × 10^5^	*R* ^2^
5 K	0.1648	1.8531	1.29	0.9956	1.2498	0.9212	3.90	0.9951
10 K	0.3036	1.8253	1.31	0.9957	1.0375	0.9516	4.95	0.9961
15 K	0.4222	2.6066	1.72	0.9949	1.8287	1.0541	2.78	0.9960
20 K	0.5104	2.1920	1.20	0.9957	1.4636	0.9325	3.74	0.9986
Average value	0.3503	2.1193	1.38	0.9955	1.3949	0.9649	3.84	0.9965

## Data Availability

Not applicable.

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
