# Peer review of "Curing Kinetics of Methylene Diphenyl Diisocyanate—Based Polyurethane Elastomers"

_polymers, 2022, doi:10.3390/polym14173525_

Round 1
Reviewer 1 Report (Previous Reviewer 1)
Comments:
1. “and only when the heating rate is show enough can we observe the difference” (line 148) – Please check the sentence
2. In figure 2, it is better to show the degree of cure percentage against time
3. How is the percentage of cure calculated under different heating rates? It is not explained in detail.
4. Please describe thee α-t relationship calculation process with data ( as shown in Figure 10) and include it in supplementary information.
Author Response
Dear editor,
Thank you for editor’ and reviewers’ opinions, these comments are very helpful to improve the quality of the manuscript. Now I response the reviewers’ comments with a point by point in revised manuscript and highlight the changes in revised manuscript. Full details of the files are listed. We sincerely hope that you find our responses and modifications satisfactory and that the manuscript is now acceptable for publication.
Reviewer #1:
- “And only when the heating rate is show enough can we observe the difference” (line 148) – Please check the sentence.
We appreciate it very much for your comments and we have rectified the expression to “The resolution and sensitivity of the DSC became lower when the heating rate increased, which resulted in the coalescence of two peaks under high heating rates” in the revised manuscript (line 150-151).
- In figure 2, it is better to show the degree of cure percentage against time.
Thank you for your kind suggestion, and figure 2 was changed to show the degree of cure percentage against time in the revised manuscript.
- How is the percentage of cure calculated under different heating rates? It is not explained in detail.
The percentage of cure was calculated by following equation: α=ΔHt/ΔHtotal
The enthalpy was calculated by integration the exothermic peak in the DSC curve and detailed explanation was added in the revised manuscript (line 90-91 and line 166-167).
- Please describe thee α-t relationship calculation process with data (as shown in Figure 10) and include it in supplementary information.
The details of the α-t relationship calculation process were shown in supplementary information.
Reviewer 2 Report (Previous Reviewer 2)
Dear Author,
Thanks for replying to my comments and incorporating the said changes. Paper is now accepted.
Author Response
Dear editor,
Thank you for your comments and recognition. A revised manuscript has been uploaded. I really appreciate your meaningful suggestions.
This manuscript is a resubmission of an earlier submission. The following is a list of the peer review reports and author responses from that submission.
Round 1
Reviewer 1 Report
The authors are requested to ask the following queries :
1) What is the novelty or contribution of your study?
2) All the models are available in literature. Why are you just following the existed models to show the cure kinetics of your prepared materials? Could you present your own developed model to show the cure kinetics?
3) Isothermal DSC is required to study the exact cure kinetics of the material.
4) What is the significance of R2 value and why the values are more than 100 in Table 4.
5) Can you show the integration calculation details of of equation 16 ?
Reviewer 2 Report
Dear Author,
Please see below my comments:
(1) What is the novelty of this work. Why MDI-based polyurethane study is needed when other isocyanates such as TDI and HDI are already studied.
(2) What are the two transitions shown in the DSC curve when the rate is 5k or 10 k. Such transition and shoulder are not observed when the heating rate is increased? Does the author has also seen TGA if they see a similar type of transitions etc?
(3) Why three model are used to describe one curing process. The author can use one model to describe the curing process.
(4) How the author ensure complete conversion of product or all reactant are consumed. FTIR study can show the conversion of isocyanate to Urethane conversion.